# Spin Dynamics of Flavoproteins

**DOI:** 10.3390/ijms24098218

**Published:** 2023-05-04

**Authors:** Jörg Matysik, Luca Gerhards, Tobias Theiss, Lisa Timmermann, Patrick Kurle-Tucholski, Guzel Musabirova, Ruonan Qin, Frank Ortmann, Ilia A. Solov’yov, Tanja Gulder

**Affiliations:** 1Institut für Analytische Chemie, Universität Leipzig, Linnéstr. 3, 04103 Leipzig, Germany; 2Institut für Physik, Carl von Ossietzky Universität Oldenburg, Carl-von Ossietzky-Str. 9-11, 26129 Oldenburg, Germany; 3Institut für Organische Chemie, Universität Leipzig, Johannisallee 29, 04103 Leipzig, Germany; 4TUM School of Natural Sciences, Technische Universität München, Lichtenbergstr. 4, 85748 Garching, Germany; 5Research Center for Neurosensory Science, Carl von Ossietzky Universität Oldenburg, Carl-von-Ossietzky-Str. 9-11, 26129 Oldenburg, Germany; 6Center for Nanoscale Dynamics (CENAD), Carl von Ossietzky Universität Oldenburg, Carl-von-Ossietzky-Str. 9-11, 26129 Oldenburg, Germany

**Keywords:** flavin, photo-CIDNP, magnetoreception, flavoproteins

## Abstract

This short review reports the surprising phenomenon of nuclear hyperpolarization occurring in chemical reactions, which is called CIDNP (chemically induced dynamic nuclear polarization) or photo-CIDNP if the chemical reaction is light-driven. The phenomenon occurs in both liquid and solid-state, and electron transfer systems, often carrying flavins as electron acceptors, are involved. Here, we explain the physical and chemical properties of flavins, their occurrence in spin-correlated radical pairs (SCRP) and the possible involvement of flavin-carrying SCRPs in animal magneto-reception at earth’s magnetic field.

## 1. Flavins: Electronic Properties, Reactivity, and Photochemistry

Flavin derivatives perform an outstanding role in the manifold of biological processes. These colorful compounds have unique photochemical and redox properties and, thus, possess interesting chemical and biological potential [1]. Extensive research has been conducted on flavoenzymes, which rely on a flavin cofactor. The mechanisms and structures of these enzymes have been thoroughly documented, with a particular focus on their diverse catalytic activities. The main structural feature of flavins is their reactive 7,8-dimethyl-10-alkylisoalloxazine core structure that is part of the three most important flavins in nature (-)-riboflavin (RF, **1**), flavin mononucleotide (FMN, **2**), and flavin adenine dinucleotide (FAD, **3**) (Figure 1). RF (**1**), widely known as vitamin B_2_, performs a key role in energy metabolism [2]. Furthermore, it serves as the starting material for the in vivo synthesis of **2** and **3**. They are the biologically active forms of RF (**1**) and act as rather non-covalently than covalently bound cofactors in most flavoenzymes [3,4,5,6].

Since the first studies on enzyme class, containing FMN (**2**) or FAD (**3**) as a prosthetic group, in the middle 1930s by Warburg [7], Krebs [8], and Theorel [9], it is known that flavins perform a crucial role in numerous oxidation and reduction reactions. This includes the four primary energy metabolism systems, photosynthesis, aerobic respiration, denitrification, and sulfur respiration. Unlike other cofactors, flavins are, in principle, able to handle both one-electron (single-electron transfer, SET) and two-electron transfer processes [10]. This special ability is due to the three redox states of the isoalloxazine moiety. The quinoid form **4** (Fl_ox_), which possesses a conjugated π-system between the rings, is a powerful oxidant, which constitutes the primary reactivity towards substrates. In addition, flavins exist in their radical Fl_seq_ semiquinone (**5**) and the completely reduced Fl_red_ form (**6**, Figure 1).

The typical redox potential for free RF (**1**) in solution at pH 7 is E_m_ = −200 mV. E_m_ is the so-called two-electron midpoint reduction potential and is defined as the average value of both oxidation/reduction steps. This is due to the frequently occurring low stability of the one-electron reduced state **5**, which results from the alteration of the planarity when the redox state changes [11]. However, in enzymes, the E_m_ value of RF (**1**) varies drastically from −400 mV to +60 mV, because of stabilizing effects on the individual oxidation states originating from the protein’s environment [12,13,14].

One of the most important oxidizing agents in enzymatic flavin catalysis is molecular oxygen. Due to the high reduction potential of O_2_, it is an excellent oxidant from a thermodynamic point of view [15]. Flavin-dependent oxidases and monooxygenases utilize O_2_ to perform oxygen transfer reactions, such as *Baeyer-Villiger* oxidations [16] and hydroxylation-dehydrogenations [17]. For both enzyme classes, the transformation starts with a SET from ground-state triplet O_2_ to the reduced ground-state singlet flavin **6** forming a caged radical pair **7** (Figure 2). In case of oxidases, O_2_ is used just as an electron acceptor, yielding hydrogen peroxide and the oxidized cofactor **4** after a second SET from the flavin cofactor to the superoxide. However, monooxygenases form C(4a)-hydroperoxyflavin (**8**) as the key reactive intermediate from singlet-triplet interactions. After inserting an oxygen atom in the substrate hydroxyflavin (**9**) is generated that gives oxidized flavin **4** after the loss of water [18]. Uncontrolled reduction in dioxygen associated with a spontaneous collapse of the caged radical pair or the peroxyflavin often produces toxic reactive oxygen species (ROS) as side products that may react non-specifically with cellular components and, thus, are toxic [19]. Thus, nature developed oxygen-utilizing enzymes controlling and fine-tuning the oxidation potential of oxygen [20]. The two flavoenzyme classes mentioned above react rapidly with their small-molecule oxidizer, while dehydrogenases, a group of enzymes also categorized as O_2_-responsive flavoenzymes, interact in most cases very slowly with oxygen [21]. Dehydrogenases yield a mixture of H_2_O_2_ and a very reactive superoxide anion. In general, enzymes require certain tactics to improve the rate of the reaction with oxygen. Improving accessibility to O_2_ in the protein environment and stabilizing the trapped [O_2_^−^ FlH^●^] are here among the most important tools [14].

Contrary to the above-mentioned mechanisms utilizing SET processes, certain flavoenzymes possess the ability of a two-electron transfer [22]. One example of such enzymes are flavin-dependent halogenases. They catalyze the formation of carbon-halogen bonds in particular at electron rich aromatic C-atoms by oxidizing a halide ion (Figure 3a) [23,24]. Here, reduced flavin (**6**) is directly attacked at the C4a position generating hydroperoxide species (**8**), which undergoes a nucleophilic substitution at the electrophilic oxygen atom O1 by a halide Hal^−^, such as chloride. The formed hypohalous acid HOHal (**10**) is a potent electrophilic halogenating agent. The generated hydroxylated flavin **9** produces oxidized flavin **4** that is reduced by a flavine-reductase to close the catalytic cycle [25,26,27].

In addition to their versatile behavior utilizing molecular oxygen, flavins also possess very interesting photophysical properties, which will be addressed in the following chapters in detail. The absorption peaks of Fl_ox_ (**4**) in aqueous solution show four maxima at 445, 375, 265, and 220 nm. π → π* transitions are responsible for the relatively high molar absorption coefficients [28]. In nature, these photophysical properties are utilized in light-dependent flavin catalysis for one-electron reductions (Figure 3b). In these processes, upon excitation by the light a reduced flavin anion (**6***) species transfers a single electron to a substrate. The now generated, highly reactive semiquinone **5** reduces the corresponding substrate through a second SET. One enzyme conducting this type of reaction is the DNA-photolyase, which reduces thymine dimers generated by UV-light and, thus, is part of the DNA-strain repair mechanism in cells [29]. The redox-potential of the thymine dimer in contrast to its one-electron reduced radical is −2.2 V vs. saturated calomel electrode (SCE), which underlines the high reducing potential of flavin in this case [30,31].

The remarkable photophysics of flavins are not only used by nature. Additionally, in synthetic organic chemistry, flavins perform a crucial role as photocatalysts [32], mostly in oxidative reactions [33]. As the chemical activity of flavins in a non-enzymatical surrounding is significantly reduced, chemically modified, tailored flavins are used very frequently. The repertoire of flavin catalyzed photoreactions spans from the oxidation of alkyl groups, cycloadditions of diens, dehalogenation of aryl halides, aromatic halogenations to cofactor regenerations in biocatalysis [34,35,36,37].

Another type of flavin-dependent enzymes performing an important role in the biology of animals and plants are cryptochromes. Cryptochromes are flavin-containing, blue-light absorbing proteins that are assumed to be crucial for the magnetic sense of migratory birds, the circadian clock of insects and mammalians, and for the movement and development of plants [38,39]. While the mechanism for magnetic perceptions in birds is still uncertain, a likely explanation is based on the electron-transfer chain between the acceptor flavin and the neighboring tryptophan donors in the protein environment. Starting with a photoexcited flavin in the cryptochrome, tryptophan residues in close proximity, then transfer electrons to the flavin, yielding a flavin-tryptophan radical pair. It is assumed that this radical pair is a magnetoreceptor due to its spin-dynamics and, thus, can interact with the weak magnetic field of the earth [40,41]. Dependent on how the cryptochromes are aligned, different physiological signals are generated [42]. Furthermore, the observed radical pair mechanism led to increased research of magnetic fields acting on anthropogenic biological systems, including flavoproteins [43]. Generation of a spin-correlated radical pair (SCRP) can be investigated by so-called CIDNP nuclear magnetic resonance spectroscopy (NMR) which will be discussed in the following chapters of this review.

## 2. Flavins Acting in the Classical Radical Pair Mechanism in Liquid-State NMR

The flourishing field of spin-chemistry is based on a bold assumption very much contrary to chemical intuition: Even if the recombination of two radicals is energetically highly favorable, it will not happen if it is “spin forbidden”. Hence, recombination, although thermodynamically allowed, is kinetically controlled by the electronic spin states. In this chapter, we will report on the discovery and the implications of this principle. We will see that flavins are ideal “players” in the field of spin-chemistry.

### 2.1. Discovery of the Principle of Photo-CIDNP in Liquid-State NMR

The discovery of the photo-CIDNP effect originated from the 1967 research of Bargon, Fischer and Johnsen [44]. It was observed during the thermal decomposition of organic peroxides and azo compounds in a series of NMR experiments. As the authors noted, intermittently, in some reactions, the proton resonance lines of the reaction products appeared in emission (i.e., negative) rather than in enhanced absorption (i.e., positive). This allowed them to suggest that the molecules were originally formed in a negative spin polarization state of their proton spin system due to the chemical reaction. The same observation has been made independently by Ward and Lawler for the reactions of alkyl halides with lithium alkyls afterward [45]. It then was discovered that nuclear polarization could also be induced by photoexcitation of a redox-competent dye in the presence of the biomolecule of interest 1968 [46] and the effect has been coined photo-CIDNP. However, the first observation of this phenomenon [47,48] appeared already in 1963, since radical chemical reactions may give rise to an analog effect in electron paramagnetic resonance (EPR) spectra, but the observation was not further discussed. Therefore, the first work cited as CIDEP (chemically induced dynamic electron polarization) was that of Paul and Fischer, who, in 1970 [49], studied the reduction and oxidation of propionic and isobutyric acids.

It was necessary to have a theoretical model in order to explain all the observed phenomena. The original model by Bargon and Fischer [45,50,51] proposed to explain signal amplification on the spectrum was based on an Overhauser effect and depended on electron-nuclear cross-relaxation in free radicals. However, for a variety of reasons [44,52,53,54,55], this model has now been abandoned. Kaptein and Oosterhoff [56] and Closs and Closs [54] proposed the radical pair mechanism (RPM) that satisfactorily explains large enhancements and multiplet spectra and allows for spectral simulations. To describe CIDEP spectra, this theoretical concept was adjusted, and the term “spin-correlated radical pair” was coined [57,58]. In flavoproteins, such as cryptochrome, a SCRP has also been observed (see Figure 4). For photo-CIDNP NMR experiments, SCRPs are often produced by illumination of flavin compounds, forming a molecular triplet state and, subsequently, oxidizing aromatic compounds as, e.g., an aromatic amino acid. In this case, due to spin conservation, the SCRP is born in its triplet state.

According to the model, nuclear spin states can control the electronic state and therefore the chemical fate [60]. The central elements of this mechanism are two radicals (molecules with an unpaired electron) forming a SCRP, which can exist in its singlet $\left|S\right.\rangle$ or one of the three triplet states $\left|T_{0}\rangle \right.,\;\left|T_{+}\rangle \right.,\;\left|T_{-}\rangle \right.$ (Figure 5A) or in a coherent superposition (in the high-field limit) of $c_{S}\left|S\right.\rangle +c_{T}\left|T_{0}\rangle \right.\;$(Figure 5B) [61]. In many cases, the recombination reaction of a singlet state SCRP occurs efficiently, while the recombination of the triplet SCRP is kinetically spin-forbidden.

Since the recombination of the SCRP reaction is spin-selective, the singlet-triplet inter-conversion affects the recombination rate (Figure 5C). For a SCRP formed by the radicals 1 and 2, and a single magnetic nucleus (*I* = 1/2) with isotropic hyperfine coupling to one of the radicals, the Hamiltonian is written as follows:(1)$$\hat{H}=\omega_{1}{\hat{S}}_{1z}+\omega_{2}{\hat{S}}_{2z}-J_{ex}\left({\hat{S}}_{1}\cdot {\hat{S}}_{2}\right)+a_{iso}{\hat{S}}_{1z}{\hat{I}}_{z}+\frac{a_{iso}}{2}\left({\hat{S}}_{1}^{+}{\hat{I}}^{-}+{\hat{S}}_{1}^{-}{\hat{I}}^{+}\right)$$
where ω is the electron Larmor frequency, $a_{iso}$ is the isotropic part of the hyperfine interaction, and J is the exchange interaction in the SCRP. The difference in the Larmor frequencies of the electrons and the isotropic part of the hyperfine interaction leads to the mixing of the |*S*⟩ and |*T*_0_⟩ states. The term $\frac{a_{iso}}{2}\left({\hat{S}}_{1}^{+}{\hat{I}}^{-}+{\hat{S}}_{1}^{-}{\hat{I}}^{+}\right)$, in which the raising (lowering) operators are defined as ${\hat{S}}_{}^{+\left(-\right)}={\hat{S}}_{x}^{}\pm i{\hat{S}}_{y}^{}$, and ${\hat{I}}_{}^{+\left(-\right)}={\hat{I}}_{x}^{}\pm i{\hat{I}}_{y}^{}$ is associated with |*S*⟩->|*T*_+_⟩ and |*S*⟩->|*T*_−_⟩ transitions, which are possible only in weak magnetic fields (fields less or about the hyperfine coupling, HFC) or in the neighborhood of the |*S*⟩ to |*T*_±_⟩ level-crossing point. This concept has been discussed in detail here [62,63,64].

Hence, since only the singlet SCRPs are allowed to recombine and form “recombination products”, the triplet radical pairs will diffuse apart and form so-called “escape products” when the time scale for diffusion is shorter than the reverse intersystem crossing time (in which otherwise a singlet could be obtained first). This selection process is called “spin sorting”. The distinguished reaction products appear to be enriched in certain spin states of magnetic nuclei, i.e., non-equilibrium polarized [65] leading to a signal pattern in the NMR spectrum, which is explained by Kaptein’s sign rules [66,67]. According to these rules, the phase of the enhancement Γ, i.e., positive or negative, for a given nucleus is provided by the product of four signs:(2)$$\Gamma =-\mu \cdot \varepsilon \cdot sgn\left(\Delta g\right)\cdot sgn\left(a_{iso}\right)$$
where $sgn\left(a_{iso}\right)$—the sign of the isotropic hyperfine coupling constant for that nucleus, $sgn\left(\Delta g\right)$—sign of the difference between the two *g*-factors of the two radicals, μ—an expression for the electronic spin state of the precursor state, and ε—the reaction product, which is positive for recombination products and negative for escape products [68]. However, it should be emphasized that the sign rules apply only when, during the diffusion convergence, the radicals of the geminate radical pair have sufficient time to undergo a singlet-triplet inter-conversion one or more times. Therefore, deviations from the Kaptein rules can be expected when the viscosity of the solution is high, or in systems in which the hyperfine interaction constants or *g*-factor differences are relatively large [65], as discussed by Salikhov [69]. For many of these ^1^H liquid-state photo-CIDNP NMR studies, flavin compounds were used for the efficient light-induced formation of the SCRP, and vice versa, photo-CIDNP NMR offered a method to detect short-lived flavin radicals in solution [70].

There also have been works on the photo-CIDNP effect beyond ^1^H liquid-state NMR using flavin dyes, including studies in the gas phase [71,72], on various nuclei as ^19^F nuclei, using various fluorescent dyes [73], and studies of organic-chemical reaction mechanisms [74,75,76]. Protein structures have been investigated using the photo-CIDNP effect [77]. Methods of time-resolved photo-CIDNP effect were developed [78], the concept of isotropic mixing by Vieth and co-workers [63,79,80] and field-cycling photo-CIDNP NMR of liquid samples [79,81,82] was developed. Very strong CIDNP effects have been found in molecular systems, such as radicals complexed with free-moving limit micelles [83,84] and biradical systems [85,86,87]. Hence, photo-CIDNP in the fluid phase with SCRPs formed by small molecules has been explored in many aspects and, in general, well rationalized in terms of the classical RPM.

### 2.2. Flavoproteins Studied by Liquid-State Photo-CIDNP NMR

As mentioned above, FMN and FAD are present in a great variety of biocatalytic processes. In particular, the common 7,8-dimethyl isoalloxazine ring structural motif is responsible for not only the color, but also electron-mediation processes and has been investigated intensively. Initially, continuous-wave (CW) EPR [88,89,90] and ENDOR [91] experiments were applied to study the electronic structure of flavin isoalloxazine moiety in its Fl_seq_ state in solution. The active function of the flavin in electron-transfer reactions and enzyme-catalyzed processes were also identified [92,93]. Moreover, protein-cofactor interactions in flavin radicals were unraveled and the formation of a molecular triplet state [94,95] led in some cases to a formation of a covalent bond with nearby amino acid residue [96].

Liquid-state photo-CIDNP NMR studies on these flavoproteins carrying both donor and acceptor were hampered by the impossibility to undergo diffusion of the two parts of the SCRP, i.e., forming escape products. In 2005, however, Richter et al. [97] observed strongly spin-hyperpolarized signals from the light-oxygen-voltage-sensing number 2 (LOV2) domain of the blue-light photoreceptor phototropin from oat (*Avena sativa*), when the ^13^C liquid-state NMR spectra were recorded under illumination (Figure 6). This study has been the first observation of a photo-CIDNP effect in an integral flavoprotein system, having both electron donor and acceptor incorporated within the protein. This new observation broke the pattern that the photo-CIDNP effect in electron-transfer proteins was limited in photosynthetic reaction centers (RCs) studied under magic-angle spinning (MAS) NMR conditions. In this highly relevant experiment, the FMN was uniformly labeled with ^13^C and the cysteine located near FMN was mutated to alanine (C450A) to allow the formation of SCRPs. Based on the observation of the strong molecular triplet populations in the system observed by optical spectroscopy and time-resolved EPR [98], Eisenreich et al. [99], in 2008, proposed a cyclic reaction scheme for the production of photo-CIDNP in LOV2-C450A of phototropin: Upon illumination of the cofactor FMN, a molecular triplet state of the FMN and, induced by electron transfer from an electron-donating aromatic amino acid, a SCRP is formed. Due to spin-conservation rules, the SCRP is formed in its triplet state.

In 2015, Eisenreich et al. [100] presented strategies for ^13^C labeling of tryptophan in LOV2-C450A of phototropin and obtained ^13^C liquid-state photo-CIDNP NMR spectra of single and double mutant LOV2 and discussed the benefit of fractional labeling method on the light-induced nuclear spin polarization quantitative analysis in terms of their strongly attenuating cross relaxation pathways. A very similar sample, phototropin LOV1-C57S, was studied by ^13^C MAS NMR under illumination and showed a strong solid-state photo-CIDNP effect (see below) [101,102,103].

Recently, Pompe et al. [104] investigated the electronic structure of flavin semiquinone radicals in solution in terms of their ^13^C hyperfine coupling constants by using the photo-CIDNP effect and discussed the spin density distribution within the radicals, the HFC constants determined in this work provide the reference for further studies on ^13^C hyperfine couplings in the transient radical involved flavoprotein system.

## 3. Flavoproteins Showing a Solid-State Photo-CIDNP Effect

While the photo-CIDNP effect in liquids relies on the classical RPM, the latter cannot be used for the explanation of the effect in the solid-state. Since diffusion of the SCRP, which is required for the classical RPM, is severely limited, spin sorting does not lead to a separation of the hyperpolarized signals. However, the observation of a photo-CIDNP effect in frozen photosynthetic RCs from *Rhodobacter sphaeroides* [105] demonstrated the possibility that the effect might also occur in the solid-state. While time-resolved photo-CIDNP MAS experiments showed that classical spin-sorting is active during the spin-evolution of the SCRP, the effect of the RPM vanishes after about 100 μs [106]; instead, steady-state nuclear hyperpolarization is observed, which is not caused by the RPM. Therefore, different mechanisms were proposed, namely, three-spin mixing (TSM) [107,108], differential decay (DD) [109], and differential relaxation (DR) [110,111]. The TSM relies on the coupling of two electron spins to a nuclear spin, which evolves coherently in the singlet-triplet manifold of the SCRP and converts electronic into nuclear hyperpolarization. It is driven by the electron-electron dipolar coupling and the anisotropic pseudosecular hyperfine coupling. These interactions lead to a degeneracy of three of the four electron-electron-nuclear energy levels in the S-T_0_ manifold and enabling mixing between the states allowing the conversion of electron-electron zero-quantum coherence into nuclear anti-phase coherence. It is maximized at the double-matching condition and vanishes if pseudosecular hyperfine coupling are zero. This generally excludes that the TSM is active in liquids; however, given that the molecule or protein is large, and tumbling is substantially slowed, anisotropic electron-nuclear interactions are not averaged allowing for photo-CIDNP contributions from TSM. The DD mechanism relies on different lifetimes of the singlet and the triplet branch in order to build up hyperpolarization. This difference breaks the symmetry of the SCRP by pseudosecular hyperfine interactions and, thus, leads to a conversion of polarization into nuclear coherence. It depends on a single-matching conditions since it does not rely on electron-electron dipolar couplings. In photosynthetic reaction centers, TSM and DD contribute simultaneously to the photo-CIDNP signals with TSM as its dominating factor leading to an emissive signal pattern. Polarization from DD; instead, it leads to a set of absorptive signal intensities. In systems, where a long-lived molecular triplet state occurs, for example, photosynthetic reaction centers without its adjacent carotenoid (R26), the electronic triplet induces paramagnetic relaxation of nuclear spins resulting in an imbalance of nuclear hyperpolarization in the electronic ground state which is described by the DR mechanism. The polarization generated in the triplet branch is removed while polarization in the singlet branch is maintained. As a result, the polarization generated by photo-CIDNP in the singlet branch, which in photosynthetic reaction centers resembles the RPM, is the only contribution towards hyperpolarized signals in the ground state. Recently, these concepts have been unified with the liquid-state photo-CIDNP effect by level-crossing (LC) and level anti-crossing (LAC) analysis [112]. When a coupling matrix element disturbs the coherence, the LC is avoided and turned into a LAC leading to a mixing of the spin states. Analysis of LACs can predict signal enhancements at different magnetic field strengths and is able to attribute it to the influence of solid-state photo-CIDNP mechanisms.

It was speculated that the solid-state photo-CIDNP effect is an intrinsic property of photosynthesis, confined to the photosynthetic apparatus [113]. However, the observation of hyperpolarized signals under illumination in the LOV1 domain of phototropin C57S in frozen solution [103] demonstrated that also other electron transfer proteins are capable to induce this effect. Field-dependent measurements of phototropin LOV1 C57S in frozen solution are shown in Figure 7 with enhanced signals from FMN and tryptophan which are involved in the electron transfer and form the SCRP. Here, similar to photosynthetic RCs [114,115], the sign of polarization strongly depends on the field.

By employing isotope enrichment, photo-CIDNP field-cycling methods became available. Figure 8 shows the field-dependent intensity of ^13^C and ^15^N enriched tryptophan inside LOV1 of phototropin and ^1^H of the residual water [101]. While these experiments were conducted in a liquid state, LC/LAC analysis showed that anisotropic interactions are responsible for the field-dependence shown because of the inverse dependence on the gyromagnetic constant. Therefore, solid-state photo-CIDNP mechanisms are active, even in liquids provided that sufficient orientation of the nucleus is given. In addition to LOV1 of phototropin, other flavoproteins showing the solid-state photo-CIDNP effect were reported showing an orientation dependence of the electron-donating amino acid [102]. All consist of a LOV domain with electron transfer from a conserved tryptophan to FMN, however, even without tryptophan, photo-CIDNP enhancement is observed, most likely coming from tyrosine(s) of the LOV domain as the electron donor.

In Figure 9, the tentative photo-cycle for phototropin LOV1-C57S is shown. After blue-light irradiation, the FMN is in an electronically excited singlet state with subsequent intersystem-crossing (ISC) to an excited molecular triplet state. In the following, the triplet state can either relax to the electronic ground state or induces an electron transfer from the tryptophan to form the SCRP in the triplet state. The SCRP evolves under electron-nuclear spin interaction at high fields in the S-T_0_ manifold where it can recombine only from the singlet branch to the ground state. Herein lies the difference to the photo-cycle of photosynthetic RCs, where the SCRP is created in the singlet state allowing direct recombination. In LOV1, on the other hand, the SCRP is born in its triplet state. In case that S-T_0_ interconversion is affected by nuclear spin states via *a*_iso_, a certain nuclear spin state will enrich in the singlet state of the SCRP. Furthermore, nuclear hyperpolarization arising during the S-T_0_ interconversion might also originate from the TSM which requires solid-state conditions. In terms of spintronics, the nuclear spins stemming from the coherent spin evolution of the SCRP could be exploited as a spin-filter. The enrichment of nuclear spin polarization can influence electron spin polarization, e.g., in spintronic devices. Therefore, light-driven flavin systems implemented into such devices might provide a method to analyze spin interactions or detect electron spins via changes in nuclear hyperpolarization.

The recombination rate of the SCRP in the triplet state is (close to) zero, since this reaction is spin forbidden, therefore only the singlet state acts as a decay channel for the SCRP to the electronic ground state. Such scheme might be interpreted as an extreme-case DD, since the two decay branches have very different kinetics (since one branch is not at all active). Alternatively, one might consider this scheme as a special case of the DR (since one branch does not contribute at all to the nuclear spin statistics in the electronic ground state). In any case, since photo-CIDNP enhancement in LOV1 depends on anisotropic interactions [101], the involvement of DD, TSM, or both is expected. Future time-resolved photo-CIDNP MAS experiments will provide more insight into the photo-CIDNP mechanism and kinetics of flavoproteins. Furthermore, as shown in Figure 9, illumination leads to the formation of Fl_seq_ (5).

## 4. Spin-Dynamics of Flavoproteins in Animal Navigation

Furthermore, flavoproteins were found to have important roles in macroscopic biological processes such as the time cycles of cicadas [117,118] or the so-called magnetoreception in migratory species [38,39,42,119,120,121,122,123,124,125]. Magnetoreception allows the migratory species to detect the Earth’s magnetic field, allowing them to use an innate magnetic compass to precisely navigate between breeding and wintering grounds [42]. This ability is remarkable considering the fact that the Earth’s magnetic field has a significantly low intensity of about ~50 μT.

The underlying possible mechanism was suggested by K. Schulten and others in 1978 [119]. They stated that an RPM is a leading hypothesis to explain the sensing of such weak magnetic fields by an organism. Here, the sense is fundamentally driven by a light-activated radical pair that can recombine only in its singlet state but reacts to an alternative product P independent of its spin state (see Figure 10A) [120,126,127,128,129]. Due to a direction change of the magnetic field with respect to the radical pair, different yields for the product formation and recombination could be realized and, thus, ultimately a differentiation of the heading direction can be achieved by the migratory organism [126]. The most promising candidate for a radical pair-based magnetoreceptor is the protein cryptochrome, which can be found for instance in the eye of the night-migratory songbird European robin (lat. *Erithacus rubecula*, ER) [39]. Within the protein, an FAD cofactor is bound adjacent to a chain of tryptophan residues (Figure 10B) [42,126,127,129,130]. This chain of molecules is thought to be the center for the formation of a SCRP.

By interaction with visible blue light, an electron in the FAD is excited which subsequently triggers an electron-transfer (ET) cascade from the tryptophan (W) residues towards the FAD resulting in a spin-correlated [W^·^^+^--FAD^·^^−^] pair (see Figure 10B) [39]. In contrast to previous examples, the SCRP is born in a singlet state due to spin conservation, as the ET cascade is rapid [131]. This radical pair may interact with a weak external magnetic field which leads to a fluctuation between the singlet and the triplet state populations. Due to the Pauli principle, recombination to the original pre-excited state is forbidden for the triplet state [125,127,132]. Hence, the ratio between recombination to the original ground state of the [W--FAD] system and a subsequent product P is thought to be the key component for sensing Earth’s magnetic field, which is involved in driving the singlet-triplet interconversion.

Mathematically, the quantum yield ϕ*_i_* for a specific reaction *i* of such a radical pair can be computed as follows [130,132,133,134]:(3)$$\phi_{i}=k_{i}\int_{0}^{t^{\prime }}Tr\left({\hat{P}}_{i}\;\hat{\rho }\left(t\right)\right)dt^{\prime }$$
where $k_{i}$ is the reaction specific rate constant, $\hat{P_{i}}$ is the projection operator of a certain state, and $\hat{\rho }\left(t\right)$ is the time-dependent (*t*) density operator.

The description of the $\hat{\rho }\left(t\right)$ is governed by the Liouville-von Neumann equation [133,135]:(4)$$\frac{\hat{\rho }\left(t\right)}{dt}=-i\;\left[\hat{H},\;\hat{\rho }\left(t\right)\right]+\hat{Κ}\hat{\rho }\left(t\right)+\hat{R}\hat{\rho }\left(t\right)$$
where $\hat{H}$ is the Hamiltonian of the radical pair, $\hat{Κ}$ is a spin-selective reaction operator, and $\hat{R}$ describes the induced spin relaxation [136]. The Hamiltonian of such a radical pair considers several spin interactions:(5)$$\hat{H}=\omega {\hat{S}}_{A}+\omega {\hat{S}}_{B}+\sum_{i\;\in \;\left\{A,B\right\}}\sum_{k}{\hat{S}}_{i}\cdot A_{ik}\cdot {\hat{I}}_{ik}+{\hat{S}}_{A}\cdot D\cdot {\hat{S}}_{B}-J_{ex}(2\cdot {\hat{S}}_{A}{\hat{S}}_{B}+\frac{1}{2})$$

Here, the first two terms are the Zeeman interactions of the electron spins, and the third term is the hyperfine interaction between an electron spin ${\hat{S}}_{i}$ and a nuclear spin ${\hat{I}}_{ik}$ coupled by the matrix $A_{ik}$ and the last two terms are the electron dipolar (*D*) and exchange ($J_{ex}$) couplings [60]. The solution of the equations above is non-trivial and requires a high amount of computational resources. Motivated by this issue, software packages, such as *MolSpin*, were developed to offer a toolkit to investigate all kinds of SCPR dependent problems [134].

In addition to the Earth’s magnetic field, the radical pair spins are affected by the adjacent nuclear spins via hyperfine coupling, which is typically larger than the interaction with the Earth’s field. Recent studies revealed that variations in the hyperfine interactions are drastically influencing the magnetic compass [130,132]. Figure 11 illustrates the exemplary singlet quantum yield of a SCRP with respect to the direction of the applied external magnetic field, when including 14 nuclei spins [134]. In Figure 11C, at 90°, a significant drop (“spike”) in the quantum yield can be found which has been postulated to work like a “quantum needle” in the literature [133]. This quantum needle is delivering an important aspect of the sensibility of a radical pair based magnetic compass [133].

Additionally, the dipolar coupling and exchange interaction between the two electron spins become important when their separation is below ~3.8 nm [137,138], which is certainly the case for [W^·^^+^--FAD^·^^−^] in cryptochrome.

Up to this date, the understanding and theoretical description of the [W^·^^+^--FAD^·^^−^] radical pair remains a major part of magnetoreception research. Aspects, such as multi-spin systems, due to so-called scavenger molecules (e.g., O_2_) [123,139,140,141], the influence of other residues, such as the adjacent tyrosine (Figure 10B) [142], the influence of spin-relaxation through fluctuations of the magnetic interactions due to molecular motion or otherwise [132,136], and the exploration of the precise kinetics of these complex biological systems [38,129] are subject of current research. Additionally, recent investigations demonstrated that for cryptochrome in *Drosophila* the 52 C-terminal amino acids are sufficient to facilitate magnetoreception, even though, no tryptophan chain is involved [143]. These new findings underpin the importance of further research.

Furthermore, theoretical studies with respect to the effects of nuclear polarization accumulation on the radical pair in cryptochrome were performed, which may provide a new way to investigate the dynamics of SCRPs in cryptochromes. The motivation for this study is based on the limitation of the detection sensitivity of the RPM due to the short spin-coherence lifetime of the radical pair, which may be enhanced by nuclear polarization [142]. Different nuclear spin states are favorable for specific spin-selective reaction products, which are commonly known as spin-sorting [142]. Under specific conditions, the preservation of specific nuclear spin state populations would then lead to an increased product yield, which is required for a well-functioning magnetic compass. The discussed studies provide the first insights for studying the dynamics of the radical pair within cryptochromes of migratory species via CIDNP. Photo-CIDNP could eventually allow for understanding the possible functionality of RPM in magnetoreception. Due to the challenging accessibility of cryptochrome proteins of migratory species, however, there are very limited experimental photo-CIDNP investigations performed directly on such cryptochromes and only a few model systems were studied in recent times [40,41].

## 5. Outlook

Nature has chosen flavin systems for spin-chemical processes. Therefore, flavins provide an excellent starting position to develop both, artificial magneto-detection and light-pumped hyperpolarized NMR. Trusting the *Einheit der Natur*, as Carl Friedrich von Weizsäcker stated (*Die Einheit der Natur***: 1971,** Carl Friedrich von Weizsäcker), the authors are convinced that there will be fundamental laws of spin-physics and spin-chemistry to provide a common basic scheme. Such a common law would provide the basis for a new spin-chemistry in soft matter.

## Figures and Tables

**Figure 1 ijms-24-08218-f001:**
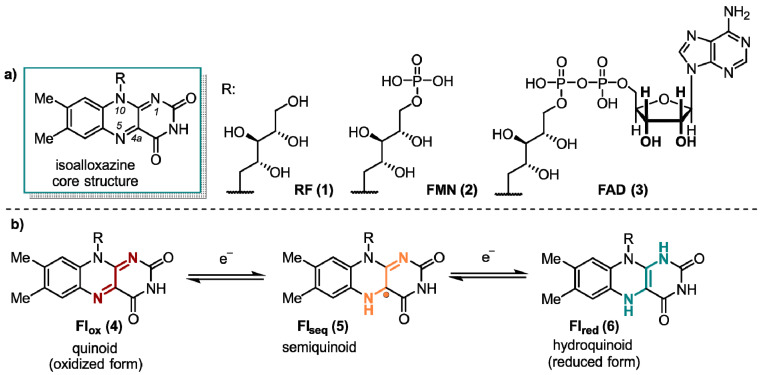
(**a**) Structural composition of the 7,8-dimethyl-10-alkylalloxazine core with its biologically most relevant derivatives all substituted at the N(10)-position. (**b**) The most common redox states of flavin derivatives.

**Figure 2 ijms-24-08218-f002:**
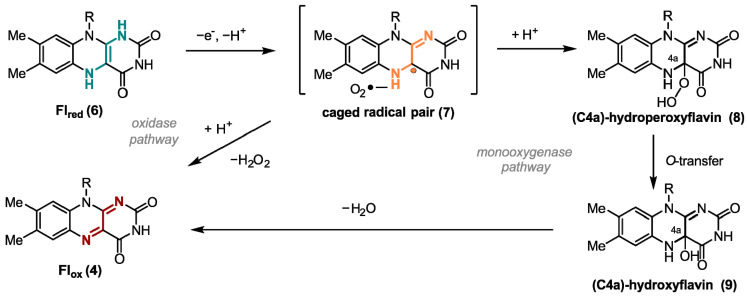
Different SET reaction pathways of the flavin cofactor.

**Figure 3 ijms-24-08218-f003:**
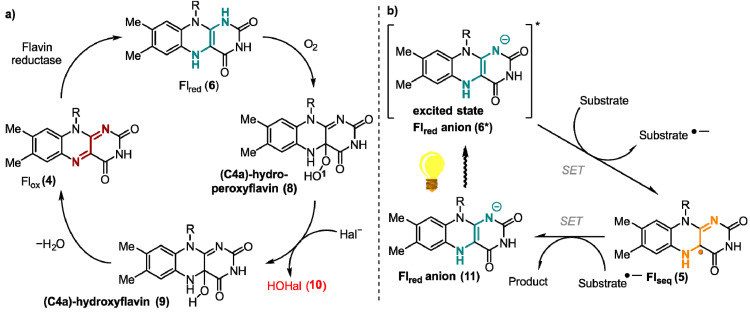
Proposed mechanism of (**a**) two-electron transfer mechanism exemplary shown for flavin-dependent halogenase and (**b**) one-electron reductions of light-induced flavin catalysis.

**Figure 4 ijms-24-08218-f004:**
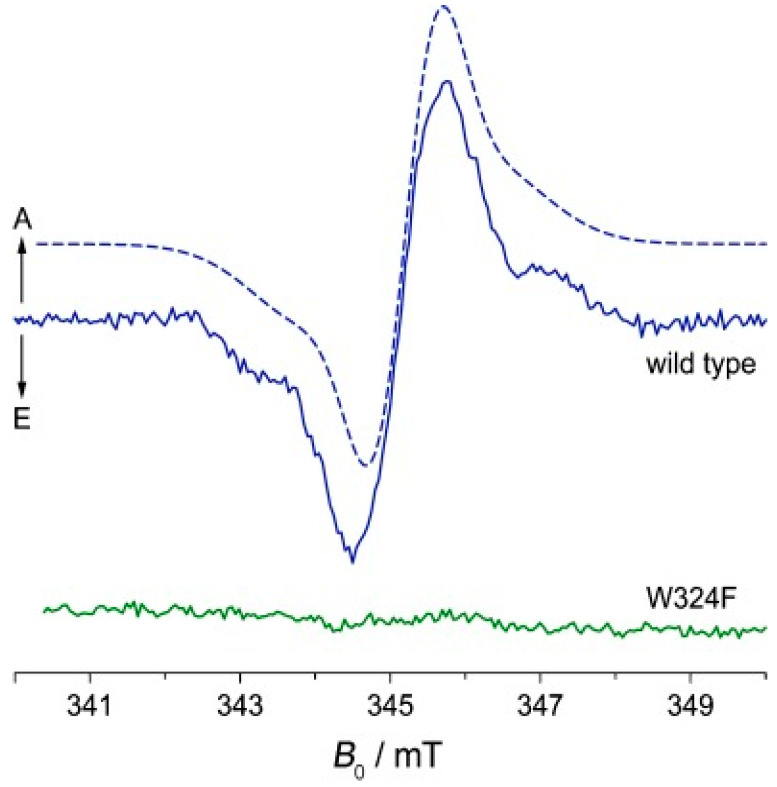
Time-resolved EPR spectrum of a SCRP in WT cryptochrome (Cry-DASH from *Xenopus laevis*) in blue and after mutation of tryptophan 324 to phenylalanine 500 ns after pulser laser excitation [59].

**Figure 5 ijms-24-08218-f005:**
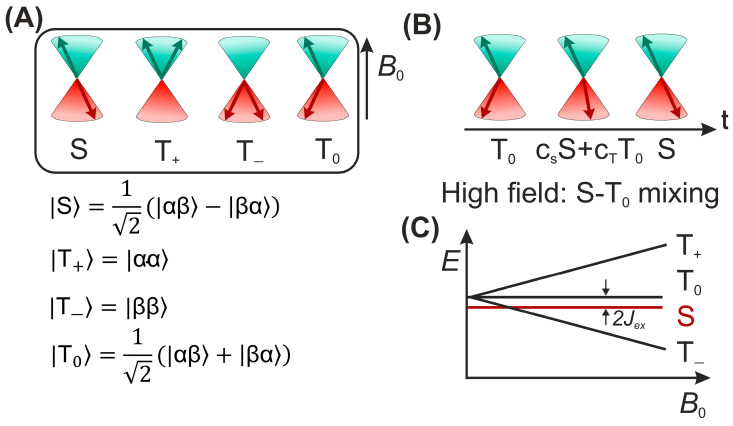
(**A**) The vector representation of four spin states of SCRP; (**B**) Vector model for visualization of the intercombination conversion of a radical pair in high magnetic fields. Left: State $\left|T_{0}\rangle \right.$; center: superposition of states; right: singlet state; (**C**) The energy of the four spin states of a radical pair along a magnetic field; $J_{ex}$ refers to the exchange coupling constant.

**Figure 6 ijms-24-08218-f006:**
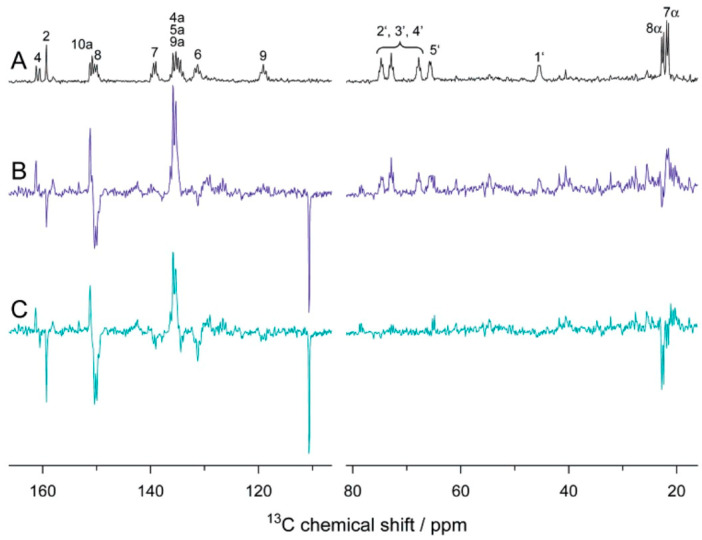
^13^C NMR spectra of [u^13^C_17_]FMN-reconstituted LOV2-C450A domain. (**A**) “Dark” spectrum. (**B**) Scaled “light” spectrum. (**C**) Difference spectrum “light”-minus-“dark” [97]. Reprinted with permission from [97]. Copyright 2005 American Chemical Society.

**Figure 7 ijms-24-08218-f007:**
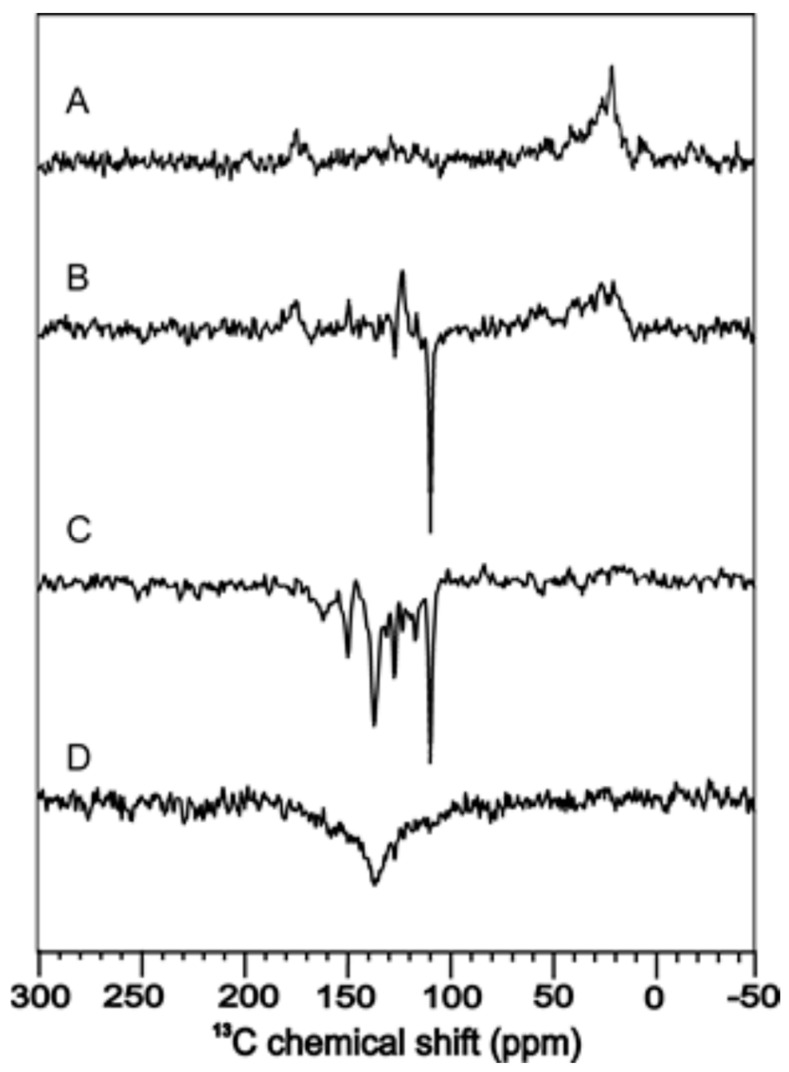
^13^C photo-CIDNP MAS NMR of phototropin LOV1 C57S at a magnetic field of 9.4 T (**A**), 4.7 T (**B**), 2.4 T (**C**), and 1.4 T (**D**) [116].

**Figure 8 ijms-24-08218-f008:**
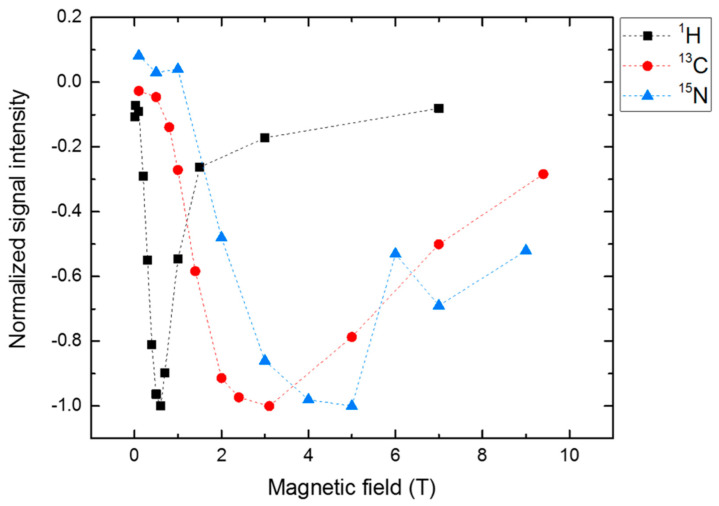
Magnetic field-dependence of ^13^C (red) and ^15^N (blue) photo-CIDNP signals from labeled tryptophan from phototropin LOV1-C57S. ^1^H (black) photo-CIDNP is observed from residual water in deuterated water. The maxima of the light-induced signals inversely correspond to the gyromagnetic constant suggesting that even in liquid, solid-state photo-CIDNP mechanisms are responsible for the signal enhancement [101].

**Figure 9 ijms-24-08218-f009:**
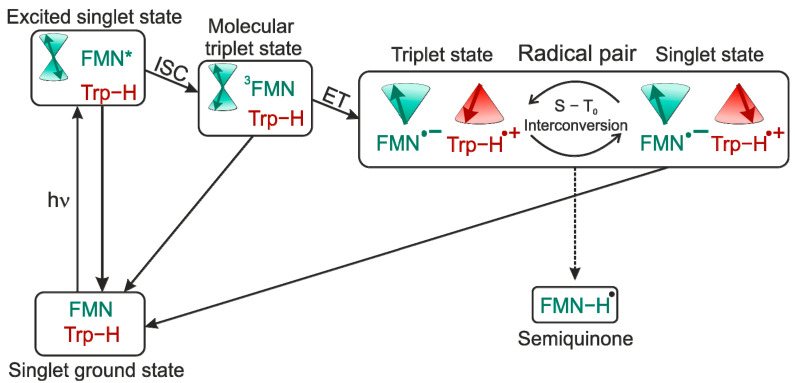
Upon blue-light illumination, the FMN is in a photo-excited state followed by a subsequent ISC to the molecular triplet state. ^3^FMN induces an electron transfer from a nearby tryptophan to form a SCRP born in the triplet state. The radical pair coherently evolves under electron-nuclear interaction between the triplet and singlet state. Since recombination from the triplet state is spin-forbidden, decay to the electronic ground-state is only possible from the singlet state. Additionally, cyclic photo-reaction process, the formation of a semiquinone-radical (Fl_seq_ (5)) with protonation of the FMN can also occur. The star indicates an excited molecular state and is not intended as an annotation.

**Figure 10 ijms-24-08218-f010:**
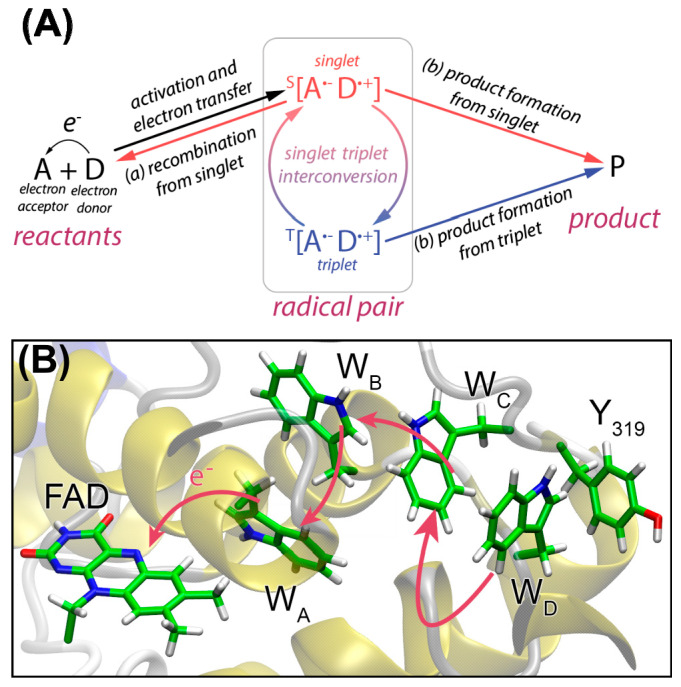
(**A**) Radical pair mechanism of electron pair from an acceptor molecule A and a donor molecule D induced by a light-activated electron transfer. The radical pair can be found in a singlet or triplet state and interconvert between both states. Both states will react into a product P, but only the singlet state can additionally recombine to the ground state before the electron transfer. (**B**) Formation of an SCRP in an FAD and tryptophan (W) chain in cryptochrome 4 via light-induced electron transfer cascade.

**Figure 11 ijms-24-08218-f011:**
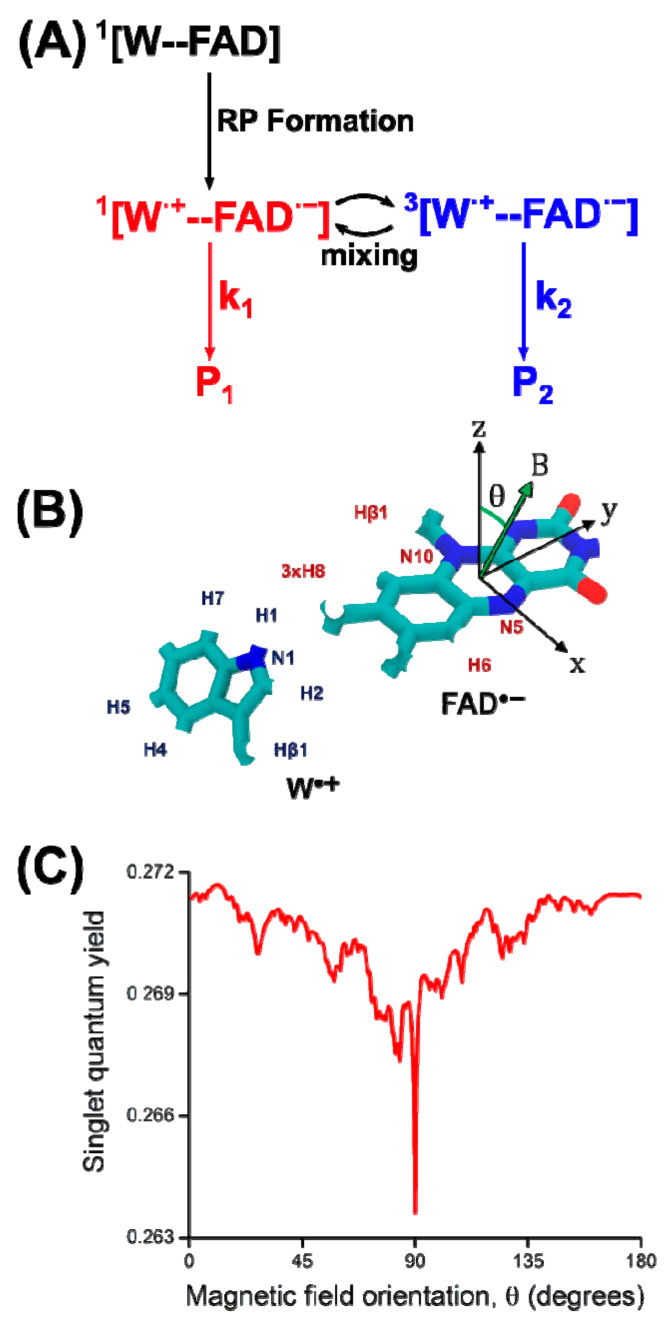
Magnetic field effects in a radical pair process. (**A**) A generic reaction scheme, where the radical pair is produced in the singlet state and interconversion between singlet and triplet states is possible. Different reactions may happen from the singlet and triplet states, leading to chemically distinct reaction products, and external magnetic fields may influence the singlet-triplet interconversion, thereby affecting the relative amount of reaction products, Φ_P1_ and Φ_P2_. (**B**) Included nuclei in the quantum yield calculation, and (**C**) magnetic field orientation concerning the two molecules. The orientation of the external magnetic field is varied in the xz-plane of the molecular reference frame as defined on the isoalloxazine moiety of FAD^•−^. (**C**) A calculation of the singlet quantum yield, Φ_P1_, for the radical pair [W^•+^--FAD^•−^] in cryptochrome with the in (**B**) coupled nuclei, adapted from [134].

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
