# Peer review of "Spin Dynamics of Flavoproteins"

_ijms, 2023, doi:10.3390/ijms24098218_

Round 1

Reviewer 1 Report

This review is an excellent educational paper describing the spin dynamics associated with radical pair formation in flavin enzymes. The reviewer has an impression that it is to update the review previously published by Evans et al. "magnetic field effects in flavins and flavoenzymes (https://royalsocietypublishing.org/doi/full/10.1098/rsfs.2013.0037)" with recent results, and presenting two new perspectives focusing on CIDNP (nuclear spin polarization) and anisotropic chemical reaction (compass effect). In this regard, it is somewhat curious that the review by Evans et al. is not cited at all. Below are a few points that caught my attention.

1) . The use of the ESR spectra of radical pairs in photosynthetic reaction centers (SCRP spectra, Figure 4) to introduce radical pairing is somewhat misleading in explaining nuclear spin polarization by the radical pairing mechanism. Indeed, we call the spectrum in Figure 4 a spin-correlated radical pair spectrum (SCRP), but its root is not the coherent mixing of S-T0 that causes CIDNP, but the interaction between spins such as J and D in the spin Hamiltonian. S-T0 mixing exists and magnetic field effects have been observed even when J and D are nearly zero, as in the radical pairs in the radiation chemistry of Molin et al. In other words, the SCRP-EPR spectra show static spin correlations, i. e. S-T energy differences (mainly 2J), and the CIDNPs are attributed to coherent spin dynamics between S-T that contribute to the recombination reaction. Therefore, it is questionable whether Figure 4 is suitable for the explanation here, and if we dare to present it, it would be better to limit it to representing the fact that radical pairs are forming and to use what has been observed with flavin enzymes such as cryptochrome.

2).  The vector model in Figure 5 is beautiful, but the spin mixing at high magnetic fields as in (B) and the energy diagram shown in (C) are likely to create confusion. For example, it is better to make J small to emphasize the strong magnetic field and show that T+ and T- are not involved. Also, the font for the and T minus states in the diagram does not match the font for T plus.  It would be better to use same fonts with the wave function bellow figure (A).

3). Conversely, notations such as|S⟩-|T+⟩ in line 198 look like wave functions and are misleading. It would be necessary to remove the ket or use an arrow instead of a hyphen.

4). As noted in the text, CIDNPs in solids are quite intricately related to the TSM, DD, and DR mechanisms. If possible, a more comprehensive and detailed review of the mechanisms of CIDNPs in solids, rather than those of CIDNPs in solution, would be more useful to the reader.

5). Minor spell missing at line454 “sradical”->”radical”

Overall, this is an excellent review and I strongly recommend that it would be published with the minor corrections noted above.

Author Response

See PDF attached.

Author Response

See PDF attached.
